# Continuous Autoregressive Modeling with Stochastic Monotonic Alignment for Speech Synthesis

**Weiwei Lin**
Department of Electrical and Electronic Engineering
The Hong Kong Polytechnic University
Hong Kong SAR, China
`weiwei.lin@connect.polyu.hk`

**Chenghan He**
Department of Computing
The Hong Kong Polytechnic University
Hong Kong SAR, China

## Abstract

We propose a novel autoregressive modeling approach for speech synthesis, combining a variational autoencoder (VAE) with a multi-modal latent space and an autoregressive model that uses Gaussian Mixture Models (GMM) as the conditional probability distribution. Unlike previous methods that rely on residual vector quantization, our model leverages continuous speech representations from the VAE's latent space, greatly simplifying the training and inference pipelines. We also introduce a stochastic monotonic alignment mechanism to enforce strict monotonic alignments. Our approach significantly outperforms the state-of-the-art autoregressive model VALL-E in both subjective and objective evaluations, achieving these results with only 10.3% of VALL-E's parameters. This demonstrates the potential of continuous speech language models as a more efficient alternative to existing quantization-based speech language models. Sample audio can be found at `https://tinyurl.com/gmm-lm-tts`.

## 1 Introduction

Transformers trained with autoregressive (AR) objectives have become the dominant approach in natural language processing (NLP) (Radford et al., 2019; Brown et al., 2020; Vaswani, 2017). The vast successes in NLP have inspired researchers to apply Transformers and autoregressive objectives to image and speech domains as well (Ramesh et al., 2021; Wang et al.; Betker, 2023). Since speech and images are continuous signals, discretization is a critical first step before applying discrete autoregressive training. As a result, autoregressive modeling of images and audio typically involves two stages of training. In the first stage, a VQ-VAE (Van Den Oord et al., 2017) or a variant of VQ-VAE (Zeghidour et al., 2021; Défossez et al., 2022; Kumar et al., 2024) is trained to encode input data into discrete latent representations using a vector quantization bottleneck. After training the VQ-VAE, an autoregressive model is trained on the discrete latent codes produced by the encoder (Ramesh et al., 2021). The AR model captures the sequential dependencies in the latent space, learning to predict the next latent code based on previous ones, which enables high-fidelity generation. One of the most attractive features of AR modeling in speech is its ability to model both the distribution of acoustic vectors and the duration simultaneously. This is particularly beneficial compared to non-autoregressive models, which often require separate duration modules and external alignments (Ren et al., 2020; 2019; Li et al., 2022). Another advantage of AR models is their capability for in-context learning of speaker style and voice through prompting, which has proven to be more powerful than traditional speaker embedding-based voice cloning (Wang et al.; Casanova et al., 2024).

It has been found that to faithfully reconstruct audio signals, the required codebook size can be prohibitively large (Zeghidour et al., 2021), and the use of discrete codes often introduces artifacts and mispronunciation during reconstruction (Casanova et al., 2024; Betker, 2023). Additionally, codebooks are often under-utilized, and training stability can be compromised due to the use of the Straight-Through (ST) estimator (Łańcucki et al., 2020; Mentzer et al., 2023). While using multiple codebooks with residual vector quantization (RVQ) mitigates some of these issues (Zeghidour et al.,

Table 1: Comparison of VALL-E and Our Method.

| Method | VALL-E | Ours |
| --- | --- | --- |
| Codec Model | RVQ | GMM-VAE |
| TTS Model | Decoder-only-Transformers | GMM-LM-Mini |
| Codec Related Param. | 16.7M | 0 |
| TTS Model Param. | 496.5M | 51.5M |
| Dominant Computation | $2DL^2$ | $DL^2$ |
| All Tokens AR | No | Yes |
| Strict Mono. Alignment | No | Yes |
| WER $\downarrow$ | 6.04 | **2.83** |
| Q-MOS $\uparrow$ | 3.54 | **4.11** |

2021; Défossez et al., 2022), it comes at the cost of requiring a specialized second-stage model to handle the additional codebooks (Borsos et al., 2023b; Wang et al.). To address the computational overhead introduced by these extra codebooks, some approaches avoid using all tokens in autoregressive modeling (Wang et al.; Lyth & King, 2024; Copet et al., 2024).

All of the aforementioned issues stem from the use of quantization during VAE training. In this paper, we demonstrate that vector quantization is not a necessary prerequisite for learning an autoregressive model. By constraining the latent representation to be a continuous multimodal distribution during VAE training and recovering it in the second stage with a continuous autoregressive model, we are able to build an autoregressive model that retains all the advantages of AR modeling (prompting, integrated duration modeling, and sampling) without many of the challenges associated with VQ. Table 1 compares our proposed continuous autoregressive model with VALL-E (Wang et al.), a state-of-the-art autoregressive text-to-speech (TTS) model. Without the need for multiple codebooks from residual vector quantization, we are able to use a single Conformer model to perform autoregressive modeling on all tokens, cutting theoretical computation in half compared to VALL-E's AR and NAR mixed approach. Furthermore, the removal of quantization allows us to build an ultra-compact TTS model with only 51.5M parameters, which is 10.3% of VALL-E's size. Despite its smaller size, our model achieves lower WER and higher MOS than VALL-E, thanks to the continuous autoregressive modeling approach.

## 2 RELATED WORK

**Neural Speech Codecs and Representations.** Unsupervised learning speech representation have long line of research including learn representating by reconstruction from a bottleneck using autoencoder (Chorowski et al., 2019; Qian et al., 2019) and mutli-taks learning (Ravanelli et al., 2020). More recently, the focus is shift to learning discrete representation, which often refer to as neural speech codecs. Neural speech codecs can be divided into two categories: semantic codecs and acoustic codecs. Semantic codecs, typically learned by clustering features from self-supervised models such as Wav2vec2, HuBERT, and WavLM (Baevski et al., 2020; Hsu et al., 2021; Chen et al., 2022), primarily preserve the phonetic information of speech (Choi et al., 2024). Due to the disentangled nature of these features, semantic codecs are often coupled with speaker embeddings for TTS (Polyak et al., 2021; Chen & Duan, 2022). Acoustic codecs, on the other hand, are designed to reconstruct the speech waveform, preserving all information from the speech, including phonetic, speaker, and acoustic environment details. Acoustic codecs are typically trained using a VQ-GAN model (Van Den Oord et al., 2017; Esser et al., 2021), which learns to reconstruct the input through a convolutional encoder-decoder architecture with GAN-based training and quantization layers. The discrete representation significantly reduces storage requirements and enhances I/O efficiency, making it an appealing alternative to traditional speech codecs (Zeghidour et al., 2021; Défossez et al., 2022) . Furthermore, it enables the direct application of language models, such as BERT and GPT (Devlin, 2018; Brown et al., 2020), to speech processing tasks. However, vector quantization can lead to mispronunciations during reconstruction (Casanova et al., 2024; Betker, 2023), and building a speech codec in this way often requires a very large codebook (Zeghidour et al., 2021). To address this, (Zeghidour et al., 2021) proposed using multiple codebooks during the vector quantization pro-

cess, where each codebook quantizes the residuals of the previous one. This approach is referred to as residual vector quantization (RVQ) in the literature. RVQ was further extended in Encodec (Défossez et al., 2022), where the authors employed multiscale spectrogram discriminators, a loss balancer, and lightweight transformers to improve both the speech quality and efficiency of RVQ. By introducing several vector quantization techniques from the image domain, along with improved loss functions, RVQ was further refined, resulting in very low-bit-rate codecs in (Kumar et al., 2024).

**Autoregressive Models in Speech Synthesis.** Tacotron (Shen et al., 2018; Wang et al., 2017) pioneered autoregressive modeling in TTS using an RNN trained with a regression loss on Mel-spectrograms. Unlike most autoregressive models, Tacotron cannot perform sampling because it relies on a regression loss. In contrast, the use of discrete speech codecs is appealing, as they can be directly applied to Transformers using standard cross-entropy loss. However, using multiple code-books from RVQ complicates the design of downstream models. Flattening all the codes leads to a quadratic increase in computational complexity with the number of codebooks. Significant ef-forts have been made to reduce computation for RVQ in downstream models, including strategies where not all tokens are used during autoregressive modeling or where different codebook codecs are modeled in separate stages (Copet et al., 2024; Wang et al.). In (Borsos et al., 2023a), the authors proposed a three-stage audio generation process with a semantic codec, a coarse acoustic codec, and a fine acoustic codec. However, using three autoregressive models results in slower inference. In (Borsos et al., 2023b), the authors proposed generating acoustic vectors in parallel across multiple codebooks by conditioning on semantic codes using a MaskGit decoding scheme. VALL-E (Wang et al.) performs autoregressive modeling on acoustic codecs by first generating the initial acoustic codecs autoregressively, then using a non-autoregressive model to predict the remaining codecs.

**Non-Autoregressive Models in Speech Synthesis.** Non-autoregressive models, utilizing adversarial learning (Kong et al., 2020; Kim et al., 2021; Lim et al., 2022), diffusion (Jeong et al., 2021; Popov et al., 2021; Huang et al., 2022; Liu et al., 2022; Li et al., 2024), and flow matching (Le et al., 2023; Mehta et al., 2024b), have become strong competitors to autoregressive models due to their faster inference speed and more stable generation. The primary challenge for non-AR models is how to align phonemes with acoustic vectors. FastSpeech (Ren et al., 2019; 2020) pioneered non-AR modeling by introducing a duration predictor and extracting phoneme durations from either an autoregressive model or an external aligner as ground truth. The duration predictor is trained with a regression loss. However, using regression loss instead of probabilistic modeling for duration has limitations. (Mehta et al., 2024a) found that probabilistic modeling produces better results than a regression-based approach, especially when modeling spontaneous speech. VITS (Kim et al., 2021) is a non-autoregressive model that supports probabilistic duration modeling using stochastic duration modeling. However, monotonic alignment search requires nested loops over the acoustic vectors and text sequences, which cannot be vectorized, severely impacting large-scale training. More recently, VoiceBox (Le et al., 2023) proposed separating the acoustic model and duration modeling, using conditional flow matching (CFM) as the objective, which leads to improved speech generation and better duration modeling.

**VAEs with Learned Prior.** While the prior distribution in VAEs is typically fixed as a standard Gaussian, numerous studies have demonstrated that learning a prior can enhance the latent space structure, thereby facilitating better representation learning. For instance, (Dilokthanakul et al., 2016) proposed using a Gaussian mixture model (GMM) as the prior for VAEs to enable unsupervised clustering, resulting in interpretable clusters and state-of-the-art performance in unsupervised tasks. Similarly, (Tomczak & Welling, 2018) introduced a method where the VAE learns a mixture of posteriors conditioned on pseudo-data as its prior. They demonstrated that this VAMP prior consistently outperformed the standard VAE across six image datasets. Additionally, (Makhzani et al., 2015) showed that instead of relying on traditional variational inference, it is possible to align the aggregated posterior with an arbitrary prior distribution using adversarial learning techniques.

# 3 AUTOREGRESSIVE MODELING WITH CONTINUOUS NEURAL SPEECH CODEC

In this section, we introduce the three major components of our model: (1) A Gaussian Mixture Models VAE (GMM-VAE) that compresses speech with a multi-modal latent constraint. (2) A Gaussian Mixture Models language model (GMM-LM) that autoregressively models the compressed

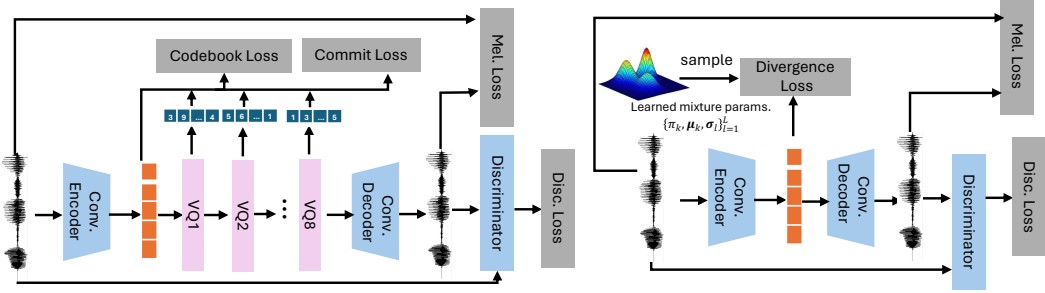

Figure 1: Training procedure and architecture difference of typical residual vector quantitation codec model (left) and our proposed GMM-VAE speech codec model (right).

acoustic vectors conditioned on text. (3) An attention mechanism that aligns the encoder and decoder in a strictly monotonic fashion.

### 3.1 LEARNING CONTINUOUS SPEECH CODECS WITH GAUSSIAN MIXTURE MODELS VAE

Learning compact representations of high-dimensional signals has been a long-standing area of research. (Hinton & Salakhutdinov, 2006; Vincent et al., 2008; Kingma & Welling, 2014; Van Den Oord et al., 2017). Variational autoencoders (VAEs) (Kingma & Welling, 2014) provide a general framework for learning generative models and performing data compression. Let $\mathbf{x}$ denote the input to the VAE and $\mathbf{z}$ the latent code. The objective is to maximize the Evidence Lower Bound (ELBO):

$$\mathcal{L} = \mathbb{E}_{q(\mathbf{z}|\mathbf{x})}[\log p(\mathbf{x}|\mathbf{z})] - \text{KL}(q(\mathbf{z}|\mathbf{x})||\mathcal{N}(\mathbf{0}, \mathbf{I})) \tag{1}$$

where $\mathbb{E}_{q(\mathbf{z}|\mathbf{x})}[\log p(\mathbf{x}|\mathbf{z})]$ is the log-likelihood of the reconstructed data, and $\text{KL}(q(\mathbf{z}|\mathbf{x})||\mathcal{N}(\mathbf{0}, \mathbf{I}))$ is the KL divergence term that encourages the latent distribution to be close to a standard Gaussian. The KL divergence acts as a critical regularization mechanism to ensure a meaningful latent space. In VQ-VAE (Van Den Oord et al., 2017), the continuous latent variable $\mathbf{z}$ is replaced by a discrete latent code $z$, corresponding to an index in a learned codebook with $K$ possible entries. Each codebook entry is a $D$-dimensional vector $\mathbf{v}$. Instead of sampling from a latent distribution during decoding, VQ-VAE quantizes the input, mapping the continuous encoder output to the nearest discrete code. Denote the output of the VAE encoder as $\mathbf{h} = E(\mathbf{x})$, and the posterior is defined as:

$$q(z = k \mid \mathbf{x}) = \begin{cases} 1, & \text{if } k = \underset{b}{\arg\min} \|\mathbf{h} - \mathbf{v}_b\|_2 \\ 0, & \text{otherwise} \end{cases}, \tag{2}$$

where $E(.)$ denotes the encoder, typically composed of several convolutional layers. Since quantization is a non-differentiable operation that prevents gradient backpropagation, a straight-through estimator is used to pass gradients from the decoder to the encoder (Van Den Oord et al., 2017). A vector quantization objective (codebook loss) and a commitment loss are added:

$$\mathcal{L} = \log p\left(\mathbf{x} \mid z_q\right) + \beta \left\|z_e - \text{sg}\left(z_q\right)\right\|_2^2 + \left\|\text{sg}\left(z_e\right) - z_q\right\|_2^2, \tag{3}$$
$$\text{where} \quad z_q(\mathbf{x}) = \mathbf{v}_k, \quad k = \text{argmin}_b \|E(\mathbf{x}) - \mathbf{v}_b\|_2$$

where sg represents the stop-gradient operator. Unlike traditional VAEs, the prior $p(z)$ is assumed to be uniform, so the KL divergence is constant and has not effect on learning. Nonetheless, regularization is achieved through vector quantization. Quantization also implicitly defines a categorical distribution during training, allowing for the second stage where an autoregressive model is trained to recover the prior/latent distribution (Van Den Oord et al., 2017). However, it has been observed that the codebook size can be prohibitively large for compact discrete audio codes (Zeghidour et al., 2021), and mispronunciations can occur during decoding (Casanova et al., 2024; Betker, 2023).

We propose a VAE compression model that supports a multi-modal latent space, but without relying on quantization. **Our key finding is that as long as the VAE stage defines a valid latent distribution that can be recovered in the second stage, quantization is not the only option.** Since our

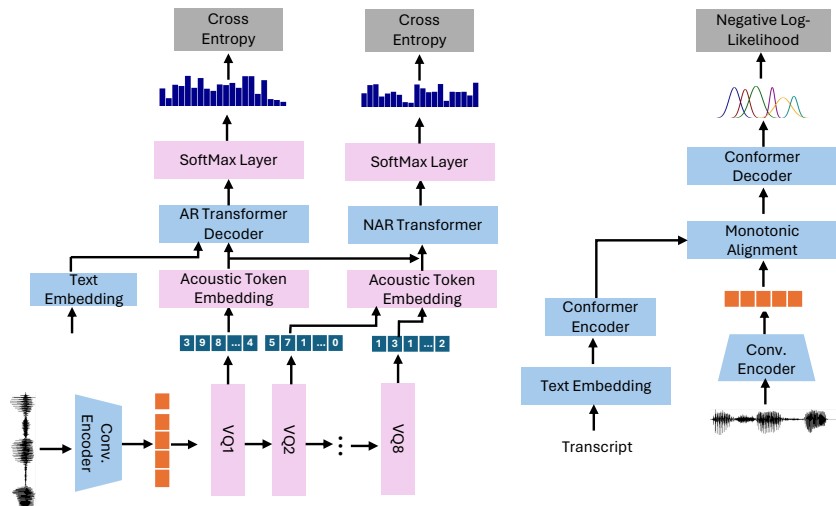

Figure 2: Training procedure and architecture difference of VALL-E with residual vector quantitation codec model (left) and our proposed GMM-LM with GMM-VAE speech codec model (right).

goal is not to build a generative model in the VAE stage, we use a deterministic mapping but skip the quantization step in Eq. 2. The posterior can be written as $q(\mathbf{h}|\mathbf{x}) = \delta(\mathbf{h} - E(\mathbf{x}))$, where $\delta$ is the Dirac delta function. The appeal of the categorical distribution defined by VQ is its ability to model multi-modal distributions, which is essential for speech. To achieve a similar effect, we use a learned mixture of Gaussians prior with parameters $\{\pi_l, \boldsymbol{\mu}_l, \boldsymbol{\sigma}_l\}_{l=1}^{L}$, allowing the continuous latent distribution to be multi-modal. The ELBO for this setup can be written as:

$$\mathcal{L} = \mathbb{E}_{q(\mathbf{h}|\mathbf{x})}[\log p(\mathbf{x}|\mathbf{h})] - \lambda \text{KL}(q(\mathbf{h}|\mathbf{x})|| \sum_{l=1}^{L} \pi_l \mathcal{N}(\boldsymbol{\mu}_l, \boldsymbol{\Sigma}_l)), \tag{4}$$

where $\lambda$ controls the strength of the regularization, $l$ is the mixture index, and $L$ is the total number of mixtures. Since our posterior is a deterministic function and the prior is a mixture of Gaussians, we no longer have an analytical solution for the KL divergence. Instead, we use Monte Carlo estimation to compute the KL divergence. Standard deep learning libraries, such as PyTorch, already support this estimation, and backpropagation through mixture sampling is derived in (Graves, 2016). We refer to a VAE trained with the proposed GMM constraint as GMM-VAE. When training GMM-VAE for speech signals, we replace the reconstruction term with Mel-spectrogram reconstruction loss and add discriminators, as done in RVQ, Encodec, and DAC (Zeghidour et al., 2021; Défossez et al., 2022; Kumar et al., 2024). Figure 1 illustrates the differences in training procedures and model architecture between the residual vector quantization model and the proposed GMM-VAE.

## 3.2 AUTOREGRESSIVE SPEECH MODELING WITH GMM-LM

Autoregressive models are typically associated with Transformer models that use discrete inputs and outputs (Vaswani, 2017; Radford et al., 2019). The conditional probabilities at time step $t$ can be expressed as:

$$P(z_t \mid z_{t-1}, z_{t-2}, \ldots, z_1, Y) = \text{Categorical}\left(f(z_{t-1}, z_{t-2}, \ldots, z_1, Y)\right) \tag{5}$$

where $f()$ is a neural network that takes in the previous predictions $z_{t-1}, z_{t-2}, \ldots, z_1$ and the conditioning information $Y$, to predict the next token. The network is typically trained with standard cross-entropy loss. However, the conditional probability of an autoregressive model does not necessarily need to follow a categorical distribution, as long as it is conditioned on the previous token and produces a valid distribution (Salimans et al., 2017).

To this end, we define an autoregressive model for the continuous random variable $\mathbf{h}_t \in \mathbb{R}^D$, where the conditional probability is represented as a mixture of Gaussians (Tschannen et al., 2024):

$$p\left(\mathbf{h}_t \mid \mathbf{h}_{t-1}, \ldots, \mathbf{h}_1, Y\right) = \sum_{n=1}^{N} \omega_n^t \mathcal{N}\left(\mathbf{h}_t; \boldsymbol{\nu}_n^t, \left(\boldsymbol{\tau}_n^t\right)^2\right) \tag{6}$$

where $\omega_n^t$, $\boldsymbol{\nu}_n^t$, and $\boldsymbol{\tau}_n^t$ represent the $n$-th mixture's weight, mean, and diagonal variances for frame $t$. $N$ is the number of Gaussian components. The mixture parameters are produced by a neural network $f()$ that takes in the previous inputs and conditioning information:

$$\left[\omega_1^t, \ldots, \omega_N^t, \boldsymbol{\nu}_1^t, \ldots, \boldsymbol{\nu}_N^t, \boldsymbol{\tau}_1^t, \ldots, \boldsymbol{\tau}_N^t\right] = f(\mathbf{h}_{t-1}, \ldots, \mathbf{h}_1, Y). \tag{7}$$

To ensure the mixture weights and variances are valid, we apply softmax to normalize the $\omega_n^t$ values and softplus to the $\boldsymbol{\tau}_n^t$ values in the network output. In the case of Gaussian mixture autoregressive modeling, we no longer need an embedding layer or a softmax layer. Since there is no stop token in the continuous case, the model requires an additional linear classifier to predict the stop token, as in (Shen et al., 2018). The autoregressive model is implemented using a Conformer encoder-decoder architecture. In this paper, we refer to the proposed Gaussian Mixture Models based autoregressive model as GMM-LM. Figure 2 illustrates the differences in training procedures and architecture between VALL-E with a residual vector quantization codec model (left) and our proposed GMM-LM with a GMM-VAE speech codec model (right).

### 3.3 STOCHASTIC HARD MONOTONIC ALIGNMENT LEARNING

In speech synthesis, monotonic alignments often result in lower word error rates and improved naturalness in generated speech(Chen et al., 2023). Our monotonic alignment mechanism is based on the work of (Raffel et al., 2017). Given the energy $e_{ij}$ between encoder state $j$ and decoder state $i$, we represent the alignment probability using a sigmoid function:

$$p_{i,j} = \mathrm{Sigmoid}\left(e_{i,j}\right) \tag{8}$$

If we use dot product to compute the energy, $p_{i,j}$ can be computed in parallel but does not enforce a monotonic constraint. To enforce monotonic attention, an iterative process (Raffel et al., 2017; He et al., 2019) is used to compute the attention score from $p_{i,j}$:

$$\alpha_{i,j} = \alpha_{i-1,j-1}\left(1 - p_{i,j-1}\right) + \alpha_{i-1,j}p_{ij} \tag{9}$$

Here, $\alpha_{i,j}$ represents the attention weight used to sum up encoder states. However, while $\alpha_{i,j}$ is monotonic in expectation, it does not guarantee that every alignment step will strictly follow the monotonic constraint for each sample. In (Raffel et al., 2017), the authors propose adding Gaussian noise to the energy term to encourage binary $p_{i,j}$ during training. However, this approach introduces bias during optimization, and discrepancy between training and test time can lead to performance issues (Chiu & Raffel, 2017). We propose replacing the soft $p_{i,j}$ with a binary value $u_{i,j}^{\mathrm{forward}}$, sampled from a Bernoulli distribution during the forward pass:

$$u_{i,j}^{\mathrm{forward}} \sim \mathrm{Bernoulli}\left(p_{i,j}\right) \tag{10}$$

This ensures that attention weights are always binary. To enable backpropagation, we approximate the gradient using a Gumbel-Softmax relaxation of the Bernoulli distribution (Jang et al., 2016):

$$u_{i,j}^{\mathrm{backward}} = \frac{\exp\left(\left(\log\left(p_{i,j}\right) + g_1\right)/s\right)}{\exp\left(\left(\log\left(p_{i,j}\right) + g_1\right)/s\right) + \exp\left(\left(\log\left(1 - p_{i,j}\right) + g_2\right)/s\right)} \tag{11}$$

where $g_1, g_2 \sim \mathrm{Gumbel}(0, 1)$, and $s$ is a temperature parameter that controls the degree of relaxation. We schedule $s$ to decrease gradually toward the end of training. Algorithm 1 provides a detailed explanation of how encoder and decoder features are monotonically aligned and fed into the decoder to compute the negative log-likelihood.

## 4 EXPERIMENT SETUP

**Training Procedure** All models were trained on the LibriLight corpus (Kahn et al., 2020), which contains 60k hours of unlabeled 16kHz audiobook speech. We transcribed the audio using Whisper

V2 (Radford et al., 2022). Since the number of utterances per speaker vary significantly, we used a class-balanced sampler to ensure each speaker was sampled the same number of times during training. For GMM-VAE, we used the same model architecture as described in the DAC paper (Kumar et al., 2024), with modifications that removed all quantization-related layers. Although it is possible to learn all the mixture parameters for GMM-VAE, in this paper, we only learned the means of each mixture. We assumed the mixture weights were equal, and the covariance matrices were set to identity. For GMM-LM, we used the Conformer encoder and decoder from SpeechBrain (Ravanelli et al., 2021) with monotonic attention modification proposed in Section 3.3. We trained models with 51.5M (Mini), 101M, 170M, and 315M (Large) parameters. The details of the GMM-LM configurations can be found in the Table A.2 in Appendix. For GMM-VAE training, we used 8 GPUs, with a total batch size of 680, and each segment length was 8960. We used scheduler-free AdamW (Defazio et al., 2024) with a learning rate of 0.001, training for a total of 1,000k steps. For GMM-LM training, we used 8 GPUs with a total batch size of 240. We used the same scheduler-free AdamW with a learning rate of 0.01 and a gradient norm clip of 0.005, training for a total of 200k steps. We did not observe any impact of random seeds on the runs.

**Evaluation Protocol** We used the LibriSpeech test set for evaluation (Panayotov et al., 2015). For zero-shot TTS, we used standard objective measures such as word error rate (WER) and speaker similarity (SIM) to assess the content and speaker fidelity of the generated speech. We used wav2vec2-large (Baevski et al., 2020), fine-tuned on LibriSpeech 960h, to transcribe the generated speech and then computed the WER against the ground truth. For SIM, we used the model released by WeSpeaker (Wang et al., 2023), trained on the VoxCeleb datasets (Nagrani et al., 2017; Chung et al., 2018). The objective metrics were computed over the entire test set. For subjective metrics, we conducted mean opinion score (MOS) evaluations on the generated audio to assess the naturalness and speaker cloning fidelity. For each model, 40 samples were generated and evaluated by 20 listeners. The listeners were asked to evaluate the speech quality and naturalness (Q-MOS) and the speaker similarity between the prompt speech and the generated speech (S-MOS). In addition to zero-shot TTS, we were also interested in the quality of GMM-VAE and GMM-LM reconstructions. We used standard reconstruction metrics such as Mel Distance, ViSQOL, and SI-SNR, as in (Kumar et al., 2024). For GMM-LM, we used teacher forcing to generate the reconstructed codec, which was then used by the GMM-VAE decoder to reconstruct the waveforms.

**Baseline Models.** We evaluated our model against state-of-the-art TTS models such as VALL-E (Wang et al.), HierSpeech++ (Lee et al., 2023) (97M Parameters), and StyleTTS2 (Li et al., 2024) (142M Parameters). For HierSpeech++ and StyleTTS2, we used the official implementations to train the models.

| Prompt Duration | Model | WER(%)↓ | SIM↑ | S-MOS↑ | Q-MOS ↑ |
|---|---|---|---|---|---|
| | GroundTruth | 2.26 | - | $4.24 \pm 0.06$ | $4.24 \pm 0.10$ |
| 3 Seconds | StyleTTS-2 | 3.32 | 0.70 | $3.44 \pm 0.10$ | $3.82 \pm 0.06$ |
| | HierSpeech++ | 3.45 | 0.71 | $3.52 \pm 0.13$ | $3.71 \pm 0.07$ |
| | VALL-E | 6.04 | 0.73 | $3.65 \pm 0.11$ | $3.54 \pm 0.08$ |
| | Ours-Mini | 2.83 | 0.80 | $3.84 \pm 0.13$ | $\mathbf{4.11} \pm \mathbf{0.08}$ |
| | Ours-Large | **2.81** | **0.85** | $\mathbf{4.04} \pm \mathbf{0.07}$ | $4.06 \pm 0.11$ |
| 8 Seconds | StyleTTS-2 | 3.34 | 0.77 | $3.57 \pm 0.07$ | $3.92 \pm 0.10$ |
| | HierSpeech++ | 3.22 | 0.75 | $3.66 \pm 0.10$ | $3.80 \pm 0.08$ |
| | VALL-E | 7.54 | 0.73 | $3.71 \pm 0.11$ | $3.58 \pm 0.08$ |
| | Ours-Mini | 3.02 | 0.81 | $3.82 \pm 0.09$ | $3.87 \pm 0.12$ |
| | Ours-Large | **2.75** | **0.88** | $\mathbf{3.92} \pm \mathbf{0.13}$ | $\mathbf{4.02} \pm \mathbf{0.10}$ |
| 15 Seconds | StyleTTS-2 | 3.02 | 0.76 | $3.52 \pm 0.13$ | $4.02 \pm 0.08$ |
| | HierSpeech++ | 3.12 | 0.77 | $3.73 \pm 0.11$ | $3.84 \pm 0.07$ |
| | VALL-E | 9.68 | 0.63 | $3.62 \pm 0.12$ | $3.32 \pm 0.12$ |
| | Ours-Mini | 2.92 | 0.80 | $3.78 \pm 0.08$ | $\mathbf{4.16} \pm \mathbf{0.09}$ |
| | Ours-Large | **2.72** | **0.91** | $\mathbf{4.17} \pm \mathbf{0.07}$ | $3.92 \pm 0.06$ |

Table 2: Comparison of models' zero-shot TTS performance on WER, SIM, S-MOS, and P-MOS across different prompt durations.

## 5 EXPERIMENT RESULTS

### 5.1 ZERO-SHOT TEXT-TO-SPEECH

We evaluated our model's zero-shot text-to-speech ability using prompts against both state-of-the-art autoregressive model VALL-E, and non-autoregressive models, StyleTTS-2 and HierSpeech++. To study the effect of prompt length on speech synthesis, we divided the prompts into three groups: 3 seconds, 8 seconds, and 15 seconds. If an utterance exceeded the specified duration, we truncated it; otherwise, we sampled another utterance from the same speaker and concatenated it until reaching the desired duration. We used two versions of our model for evaluation: a mini version with 51.2M parameters and a large version with 315M parameters, both of which output 6 Gaussian mixture parameters with diagonal covariances. Both models were trained on encoder features from a GMM-VAE model trained with 3 mixture components, constrained by $\lambda = 50$. The results are shown in Table 2. The first observation from Table 2 is that non-AR models such as StyleTTS2 and HierSpeech++ achieved much better WER than VALL-E. We found that VALL-E was prone to skipping words or failing in the middle of sentences, suggesting that simple cross-attention mechanisms may not be sufficient for autoregressive models. Contrary to our expectations, VALL-E's WER worsened as the prompt length increased. Another observation is that non-AR models do not significantly benefit from longer prompts, as there was no performance improvement from 8-second to 15-second prompts in SIM and S-MOS for HierSpeech++ and StyleTTS2. By contrast, our proposed GMM-LM model consistently outperformed both VALL-E and the state-of-the-art non-AR models in content fidelity and speaker cloning, demonstrating a clear advantage in speaker-related metrics with longer prompts. We also found that scaling our model from mini to large primarily benefited speaker-related metrics (SIM and S-MOS), while WER and general speech quality did not improve as much.

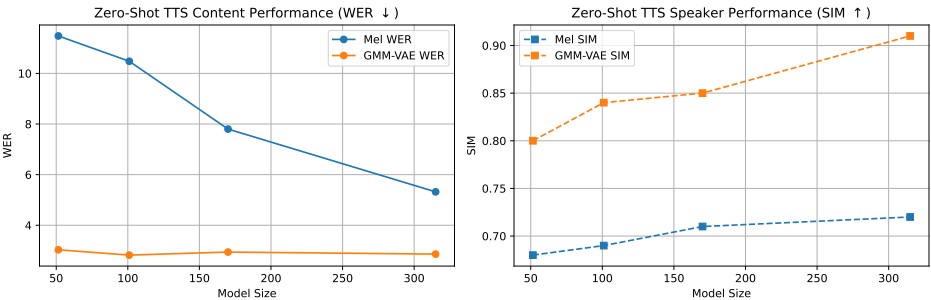

Figure 3: Comparison on GMM-LM models' zero-shot TTS performance trained on Mel-Spectrogram features and GMM-VAE features.

### 5.2 MEL-SPECTROGRAM VS. GMM-VAE FEATURES

In this section, we evaluate GMM-LM models trained on Mel-spectrograms against GMM-LM models trained on GMM-VAE encoder features. We trained models with 51.5M, 101M, 170M, and 315M parameters, each outputting 6 Gaussian mixture parameters with diagonal covariances. For models using GMM-VAE features, the GMM-VAE model was trained with 3 mixture components, constrained by $\lambda = 50$. For the Mel-spectrogram model, we used a 120-dimensional Mel-spectrogram with a hop length of 240. We evaluated the models' zero-shot TTS performance using objective metrics on the LibriSpeech test set. The results are presented in Figure 3. From Figure 3, it can be seen that GMM-LM models trained on GMM-VAE features significantly outperform those trained on Mel-spectrogram features. The discrepancy is especially pronounced with smaller models. Although increasing model size reduces the performance gap for Mel-based models, this comes at the cost of increased computation and model size. Thus, we conclude that GMM-LM with GMM-VAE features is a more efficient approach that delivers better performance for speech synthesis. Additionally, we found that GMM-LM models using Mel-spectrograms take much longer to achieve proper alignment compared to GMM-LM models using GMM-VAE features. We hypothesize that the GMM-VAE step helps extract semantic information, which greatly aids the alignment steps.

| Method | Model | Mel Distance ↓ | ViSQOL ↑ | SI-SDR ↑ |
|---|---|---|---|---|
| Encoding-Decoding | Encodec | 1.47 | 3.41 | 5.66 |
| | GMM-VAE | **1.13** | **3.96** | **8.89** |
| Teacher Forcing | VALL-E | 2.26 | 2.25 | -0.15 |
| | GMM-LM | **1.82** | **2.86** | **4.45** |

Table 3: Comparison of reconstruction quality for codec models (Encodec and GMM-VAE) and downstream models with teacher forcing (VALL-E and GMM-LM).

| No. Gaussian | Mode | $\lambda$ | | | | |
|---|---|---|---|---|---|---|
| | | 0.1 | 1 | 10 | 50 | 100 |
| 1 | GMM-VAE Recon | 0.76 | 0.89 | 1.14 | 1.67 | 2.06 |
| 1 | GMM-LM Teacher Forcing | – | 4.62 | 3.82 | 3.16 | 4.48 |
| 3 | GMM-VAE Recon. | 0.71 | 0.77 | 0.98 | **1.13** | 1.77 |
| 3 | GMM-LM Teacher Forcing | 6.72 | 3.04 | 1.87 | **1.82** | 4.27 |
| 6 | GMM-VAE Recon. | 0.73 | 0.86 | 0.95 | 1.37 | 1.83 |
| 6 | GMM-LM Teacher Forcing | 4.63 | 4.07 | 2.54 | 2.44 | 5.26 |

Table 4: Reconstruction Quality (Mel Distance ↓) of GMM-VAE and GMM-LM (with teacher forcing) across different number of Gaussians and $\lambda$ values.

## 5.3 RECONSTRUCTION QUALITY OF DISCRETE AND CONTINUOUS CODECS

In this section, we evaluate the reconstruction quality of our GMM-VAE model in comparison with the RVQ codec model, Encodec. The GMM-VAE model was trained with 3 mixture components, constrained by $\lambda = 50$. Our primary focus is on how well these models can reconstruct the input, as this greatly impacts tasks like zero-shot TTS. Even if codec models can perfectly reconstruct the input, if the downstream models are unable to predict the correct codec, the effort is futile. Therefore, we also evaluate the reconstruction quality with downstream models such as VALL-E and our GMM-LM. Specifically, we encode the input speech to generate speech codecs, then use VALL-E or GMM-LM in teacher forcing mode to predict the codec. Finally, we reconstruct the speech by decoding the teacher-forced predicted codec. We use common speech reconstruction quality metrics, including Mel Distance, ViSQOL (Hines et al., 2015), and SI-SDR. The results are presented in Table 3. It is clear from the table that our GMM-VAE significantly outperforms Encodec in speech reconstruction. The performance gap is even larger when the downstream models are included.

## 5.4 ABLATION STUDY

**Divergence constraint.** Since we no longer perform quantization when training the speech codec, the divergence constraint in Eq. 4 becomes the only factor that directly influences the latent distribution, aside from the reconstruction loss. Because our goal is to use the codec for downstream models rather than compress speech per se, we evaluate both the reconstruction quality of GMM-VAE and the downstream GMM-LM reconstruction with teacher forcing, using Mel-spectrogram distance. The GMM-LM model used is the large version with 6 Gaussian mixtures and diagonal covariances. Specifically, we investigate how the value of $\lambda$ and the number of Gaussian means in the divergence constraint affect reconstruction. The results are presented in Table 4. It is important to note that without any divergence constraint, we were unable to successfully train the downstream GMM-LM models. As expected, the VAE reconstruction Mel distance increases with larger $\lambda$. However, the effect of $\lambda$ on downstream model reconstruction is less straightforward. With $\lambda = 0.1$, although we observed very low Mel distance, we were unable to train downstream models (the model could not produce intelligible speech conditioned on text). When increasing $\lambda$ from 1 to 50, although the GMM-VAE reconstruction quality decreases, the downstream model's teacher forcing reconstruction quality improves. In terms of the number of Gaussians in the constraint term, we found

that using three Gaussians yields better results than using a single Gaussian, with both GMM-VAE and GMM-LM showing smaller Mel distances. However, increasing the number of Gaussians to 6 led to a general decrease in reconstruction quality. Since our focus is on improving the GMM-LM generation quality, we selected three Gaussians with $\lambda = 50$ as the optimal hyper-parameters.

**GMM-LM Parameterization.** The GMM parameterization in GMM-LM directly affects the output distribution. We aim to investigate how the number of Gaussian mixtures and the choice between full or diagonal covariance matrices impact model performance. We assess the generated speech using objective metrics such as WER and SIM. Furthermore, since GMM parameterization also influences the sampling process, we measure the diversity of the generated samples under different parameterization schemes. Specifically, given the target text and speech prompt, we randomly sample from the parameterized GMM to generate speech three times. We ask human evaluators to rate the diversity of the three samples on a scale from 0 to 5. These results are compared with VALL-E, StyleTTS2, and HierSpeech++, as presented in Table 5. For comparison, we also included Tacotron's approach, shown in the first row of Table 5, which trains an autoregressive model using only the mean and a regression loss. This approach does not allow for sampling. We found that using a single Gaussian with an L1 loss led to weaker performance compared to models with multiple mixtures. As we increased the mixture count to 3, and then to 6, we observed improved diversity in both diagonal and full covariance models. However, the WER improvement was mixed: generally the full covariance model worsened with more mixtures, while the diagonal covariance model maintained approximately the same WER. Finally, we did not observe better results with 10 mixtures. In conclusion, there is no clear advantage in using full covariance matrices or increasing the number of mixtures beyond 6. When comparing the speech diversity of GMM-LM with VALL-E, StyleTTS2, and HierSpeech++, our model consistently produced significantly more diverse samples. For StyleTTS2 and HierSpeech++, the multiple generated samples showed little variation. For VALL-E, while some degree of diversity were observed, the model frequently failed during random sampling. When excluding failed samples, the diversity remained limited. In contrast, our model consistently generated samples with diverse speaking styles.

| No. Mixture | Variance Type | Loss | WER ↓ | SIM ↑ | Diversity ↑ |
|---|---|---|---|---|---|
| | None | L1 Mel-Dist | 4.29 | 0.75 | - |
| 1 | Diagonal | NLL | 3.34 | 0.81 | 2.17 |
| | Full | NLL | 3.45 | 0.82 | 2.32 |
| 3 | Diagonal | NLL | 2.89 | 0.85 | 3.14 |
| | Full | NLL | 3.74 | 0.72 | 2.85 |
| 6 | Diagonal | NLL | **2.72** | **0.91** | **3.42** |
| | Full | NLL | 4.82 | 0.69 | 3.01 |
| 10 | Diagonal | NLL | 5.21 | 0.71 | 3.08 |
| **VALL-E** | | - | 9.68 | 0.63 | 1.47 |
| **StyleTTS2** | | - | 3.02 | 0.76 | 0.6 |
| **HierSpeech++** | | - | 3.12 | 0.77 | 0.3 |

Table 5: Zero-shot TTS performance of GMM-LM with different mixture counts and variance types, evaluated on WER, SIM, and sample diversity.

# 6 CONCLUSION

We presented a new autoregressive modeling approach for speech synthesis, utilizing a Gaussian Mixture VAE (GMM-VAE) for compression and a Gaussian Mixture Model language model (GMM-LM) for autoregressive generation. By eliminating the need for quantization and introducing stochastic monotonic alignment, our method ensures stable, efficient, and high-quality speech synthesis. Our experiments showed that this approach outperforms traditional VQ-based methods, achieving lower WER and higher MOS scores with reduced model complexity. These results demonstrate the potential of continuous latent spaces for autoregressive modeling in speech synthesis, offering a simpler and more effective alternative to VQ models.

# 7 ACKNOWLEDGMENT

This work was supported by the RGC of Hong Kong SAR, Grant No. PolyU 15228223. We would like to thank Prof. Man-Wai Mak for the leadership in this project.

# 8 REPRODUCIBILITY STATEMENT

We have provided detailed descriptions of the hyperparameters and training procedures in Section 4. Our implementations are based on publicly available codebases (descript-audio-codec and Speech-Brain) and datasets. With the detailed explanations of the training process and hyperparameters, readers should be able to easily reproduce our models. Additionally, we plan to release the code and pre-trained models soon.

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

# A  APPENDIX

---

**Algorithm 1** GMM-LM Monotonic Alignment and Decoder Training Procedure

---

**Require:** Acoustic features $\{\mathbf{h}_i\}_{i=1}^{I}$ extracted from the GMM-VAE encoder. Phoneme features $\mathbf{Y} = \{\mathbf{y}_j\}_{j=1}^{J}$ from the GMM-LM text encoder, and the GMM-LM Decoder $D$.

1: Compute energy terms in parallel for all $i, j$ using the dot product:

$$e_{i,j} = \mathbf{h}_i^\top \mathbf{y}_j$$

2: Sample initial alignments for all $i, j$ from a Bernoulli distribution:

$$u_{i,j}^{\text{forward}} \sim \text{Bernoulli}(\text{sigmoid}(e_{i,j}))$$

3: Save the Gumbel-Softmax relaxation $u_{i,j}^{\text{backward}}$ (Eq. 11) for gradient computation.
4: Initialize $\boldsymbol{\alpha}_0 = [1, 0, 0, \ldots, 0] \in \mathbb{R}^J$, $\mathcal{L}_{\text{NLL}} = 0$, and $\mathbf{C}_0 = []$ (an empty sequence for accumulating contexts).
5: **for** each time step $i = 1$ to $I - 1$ **do**
6:     Let $\mathbf{u}_i = [u_{i,1}, u_{i,2}, \ldots, u_{i,J}]$.
7:     Refine alignments to enforce monotonicity:

$$\boldsymbol{\alpha}_i = \boldsymbol{\alpha}_{i-1} \cdot \mathbf{u}_i + \text{shift}\left(\boldsymbol{\alpha}_{i-1} \cdot (1 - \mathbf{u}_i), 1\right)$$

    ▷ shift$(\cdot, 1)$ shifts the vector by one position to the right, filling the leftmost entry with zero.
8:     Compute the attention context using the refined alignments:

$$\mathbf{c}_i = \mathbf{Y}^\top \boldsymbol{\alpha}_i + \mathbf{h}_i$$

9:     Update the accumulated attention contexts:

$$\mathbf{C}_i = \text{concat}(\mathbf{C}_{i-1}, \mathbf{c}_i)$$

10:     Pass the accumulated context $\mathbf{C}_i$ to the decoder network $D$ to obtain $N$ mixtures parameters:

$$D(\mathbf{C}_i) = \left[\omega_1^{t+1}, \ldots, \omega_N^{t+1}, \boldsymbol{\nu}_1^{t+1}, \ldots, \boldsymbol{\nu}_N^{t+1}, \boldsymbol{\tau}_1^{t+1}, \ldots, \boldsymbol{\tau}_N^{t+1}\right]$$

    ▷ $\omega_n^{i+1}$ are normalized using the softmax and $\tau_n^{i+1}$ are passed through the softplus to ensure they are positive.
11:     Compute the negative log-likelihood for the next frame:

$$\mathcal{L}_{\text{NLL}} \mathrel{+}= -\log p\left(\mathbf{h}_{i+1} \mid \{\omega_n^{i+1}, \boldsymbol{\nu}_n^{i+1}, \boldsymbol{\tau}_n^{i+1}\}_{n=1}^{N}\right)$$

12: **end for**

---

## A.1 ABLATION STUDY ON ALIGNMENT MODULE

Even with powerful neural codecs, we found that monotonic alignment forcing remains crucial for the stability of autoregressive models. We compared our monotonic alignment approach with using direct cross-attention, location-sensitive attention (Shen et al., 2018), and various monotonic attention variants. It is worth noting that location-sensitive attention is not strictly monotonic, meaning it can still result in word repetition and lookahead errors. We also found that it was slow to train because energy terms are computed iteratively. Since the alignment module primarily affects phonetic content, we used WER to measure the performance of different alignment methods. For monotonic attention, we explored the use of pre-sigmoid noise, Gumbel Softmax, and the proposed Straight-Through (ST) Gumbel in Eq. 10 and Eq. 11. The results are presented in Table 6. As seen from the table, using cross-attention alone led to poor results—some samples even failed to produce intelligible speech. Location-sensitive attention performed better, but it still suffered from frequent word repetition errors, and both training and inference were slow. Our proposed monotonic alignment with ST Gumbel outperformed all other methods, including the original monotonic attention with pre-sigmoid noise.

| Alignment | Cross Att. | Loc. Att. | Monotonic Att. | | |
|---|---|---|---|---|---|
| | | | w. Noise | w. Gumbel | w. ST Gumbel |
| **WER(%)** | 6.6 | 6.02 | 3.77 | 3.34 | **2.72** |

Table 6: Word Error Rate (WER) comparison of various alignment methods

## A.2 DETAILS OF MODEL ARCHITECTURE

For GMM-LM, we used the Conformer encoder and decoder from SpeechBrain (Ravanelli et al., 2021) with monotonic attention modification proposed in Section 3.3. We trained models with 51.5M (Mini), 101M, 170M, and 315M (Large) parameters. The details of the GMM-LM configurations can be found in the Table A.2.

| Name | Mini | 101M | 170M | Large |
|---|---|---|---|---|
| **Encoder** | | | | |
| **No. Layers** | 3 | 3 | 3 | 3 |
| **Encoder Dim.** | 512 | 512 | 512 | 512 |
| **Feedforward Dim.** | 1024 | 1024 | 1024 | 1024 |
| **Decoder** | | | | |
| **No. Layers** | 6 | 6 | 6 | 12 |
| **Decoder Dim.** | 512 | 768 | 1024 | 1024 |
| **Feedforward Dim.** | 2048 | 3144 | 4096 | 4096 |
| **Total Params. (M)** | 51.5 | 101 | 170 | 315 |

Table 7: Model Architecture and Parameters

## A.3 RECONSTRUCTION QUALITY ON TRAINING SET

Table 4 shows that the GMM-VAE with 3 mixtures and $\lambda = 50$ achieves the best reconstruction quality on the evaluation set. To further investigate the underperformance of the 6-mixture model, we selected a subset of the training data (randomly sampled 200 speakers with 5 utterances each) and computed the reconstruction loss. The results, along with the evaluation set reconstruction loss, are presented in Table 9. From Table 9, we observed that the reconstruction loss on the training set is consistently lower than that on the evaluation set across all configurations. This discrepancy may be attributed to a slight domain difference between the LibriSpeech evaluation set and the LibriSpeech-Light training set, with the latter containing noisier recordings. Another key observation is that the 6-mixture model achieves a lower reconstruction loss on the training set compared to the 3-mixture model, yet it performs slightly worse in terms of reconstruction quality on the evaluation set. This suggests that the 6-mixture model may suffer from overfitting.

## A.4 ZERO-SHOT VOICE CLONE NOISE ROBUSTNESS

In this section, we evaluate the robustness of TTS zero-shot voice cloning against noise. Specifically, we added varying levels of noise to the LibriSpeech clean test set and used the noisy audio as prompt input for the TTS models. The generated speech was assessed using WER and SIM metrics under different noise levels, with the results presented in Table 8. From the table, we observe that, with the exception of VALL-E, most models demonstrate robustness to noisy prompts in terms of WER. However, speaker similarity (SIM) significantly decreases for models like StyleTTS-2 and HierSpeech+++. In contrast, the proposed method shows strong robustness to noise for both WER and SIM, particularly in its large version, where WER and SIM degrade only slightly across different noise levels.

| Model | WER (%) | | | SIM | | |
|---|---|---|---|---|---|---|
| | -20 dB | -15 dB | -10 dB | -20 dB | -15 dB | -10 dB |
| StyleTTS-2 | 3.45 | 3.49 | 3.54 | 0.72 | 0.71 | 0.68 |
| HierSpeech+++ | 3.61 | 3.72 | 3.91 | 0.69 | 0.64 | 0.58 |
| VALL-E | 9.84 | 10.86 | 12.61 | 0.61 | 0.54 | 0.51 |
| Ours-Mini | 2.85 | 3.01 | 3.04 | 0.77 | 0.74 | 0.74 |
| Ours-Large | **2.77** | **2.82** | **2.97** | **0.88** | **0.87** | **0.85** |

Table 8: WER and SIM comparison across models under different prompt noise levels. Lower WER and higher SIM indicate better performance.

## A.5 GMM-VAE MODEL DETAILS

The GMM-VAE architecture is inspired by the DAC model Kumar et al. (2024) and comprises a convolutional encoder and decoder. Both the encoder and decoder feature a convolutional layer, which performs upsampling or downsampling based on the stride, followed by a residual block. Each residual block combines convolutional layers with non-linear Snake activations. The encoder progressively downsamples the input audio waveform with strides of [2, 4, 8, 8], while the decoder mirrors this by upsampling at corresponding rates of [8, 8, 4, 2]. The dimensionality of the encoder and decoder is configured to 64 and 1536, respectively. The GMM-VAE contains a total of 76.5M parameters, distributed as 22.4M in the encoder and 54.1M in the decoder.

## A.6 COMPARISON OF GMM-LM AND VQ-BASED LM WITH MONOTONIC ALIGNMENT

As shown in Table 6, the proposed monotonic alignment significantly improves the WER of the GMM-LM model. This raises a natural question: can discrete AR models benefit from the proposed monotonic alignment in a similar way? To investigate this, we conducted a comparison using discrete autoregressive (AR) models with the proposed monotonic alignment method, implemented using the same architecture as the GMM-LM large version (315m), except for the discrete codec embedding layer and softmax layers. Specifically, we implemented two discrete AR encoder-decoder models:

- **Discrete AR**. The first model used discrete codes extracted from a VQ-VAE codec model, implemented with a single codebook containing 8192 entries and using the same architecture as the DAC model (Kumar et al., 2024). This model was trained to predict the next tokens using standard cross-entropy with a single softmax layer.

- **Discrete AR with Delay Pred**. The second model used discrete codes extracted from the DAC model with 8 codebooks, each containing 1024 entries. The model adopted delayed codebook prediction with multiple softmax layers, as proposed in MusicGen (Copet et al., 2024) and (Lyth & King, 2024).

The results of these experiments are presented in Table 10 of the revised manuscript and summarized below. The results clearly show that monotonic alignment contributes to WER improvement. However, even with monotonic alignment, the discrete models do not achieve the same performance

| No. Gaussian | Mode | $\lambda$ | | | | |
|---|---|---|---|---|---|---|
| | | 0.1 | 1 | 10 | 50 | 100 |
| 1 | Training Set | 0.72 | 0.78 | 0.98 | 1.54 | 1.85 |
| 1 | Evaluation Set | 0.76 | 0.89 | 1.14 | 1.67 | 2.06 |
| 3 | Training Set | 0.68 | 0.70 | 0.93 | 1.11 | 1.54 |
| 3 | Evaluation Set | 0.71 | 0.77 | 0.98 | 1.13 | 1.77 |
| 6 | Training Set | 0.67 | 0.73 | 0.86 | 1.02 | 1.32 |
| 6 | Evaluation Set | 0.73 | 0.86 | 0.95 | 1.37 | 1.83 |

Table 9: Reconstruction Quality (Mel Distance ↓) of GMM-VAE and GMM-LM (with teacher forcing) across different number of Gaussians and $\lambda$ values.

| Model | Output | Codec Model | Cross Att. | Mono. Align. |
|---|---|---|---|---|
| Discrete AR | Logits Group | VQ-VAE | 8.02 | 5.35 |
| Discrete AR with Delay Pred. | Multiple Logits Groups | DAC | 7.83 | 5.87 |
| GMM-LM | GMM Parameters | GMM-VAE | 6.60 | 2.72 |

Table 10: WER (%) of the models using different codec models, with and without the proposed monotonic alignment on LibriSpeech-test-clean.

as the proposed GMM-based version. We believe this performance gap can be attributed to two factors:

1. Quantization and discrete representations introduce limitations: Discrete representations can result in mispronunciations and artifacts. This is why some researchers use embeddings before the AR model's softmax layer as input to waveform decoders as a workaround (Casanova et al., 2024; Betker, 2023).

2. Challenges of applying monotonic alignment to RVQ-based models. RVQ models increase the complexity of TTS systems, as they require multiple softmax heads to predict codes from different codebooks. While it is possible to predict all codebooks in parallel at each timestep, this approach ignores dependencies between codebooks and yields suboptimal results. A better approach, as used in VALL-E and the delayed prediction model in Table 10, is to predict "coarse" codes first, followed by "fine" codes. However, this approach complicates monotonic alignment because alignment must be performed using only the coarse codes at each timestep. This limitation likely contributes to the higher WER observed with RVQ-based models.

