# OpenReview forum: "Continuous Autoregressive Modeling with Stochastic Monotonic Alignment for Speech Synthesis"
_ICLR.cc/2025/Conference — ICLR 2025 Poster_

### Official Review · Reviewer_UrKc · 2024-10-16

**Soundness:** 3
**Presentation:** 3
**Contribution:** 3
**Rating:** 6
**Confidence:** 4

**Summary:**

This paper applies a VAE with GMM before extracting latent representations, and then trains an autoregressive model on the extracted continuous latent. The approach models the autoregressive conditional distribution also by GMM.

**Strengths:**

The idea of using GMM-VAE to regularize the latent distribution, serving a similar role as discretization is novel and interesting. The method is also easy to understand and straightforward. The paper can also inspire research on the use of continuous variational approaches in the speech synthesis domain, which has been recently dominated by discrete-based approaches.

**Weaknesses:**

To me, the main issue of the paper is that the contribution of monotonic alignment and GMM-VAE are not separated. Specifically, the paper claims that "Despite its smaller size, our model achieves lower WER and higher MOS than VALL-E, thanks to the continuous autoregressive modeling approach." in lines 82-83. However, in experiments of Section 5, you are comparing your method with monotonic alignment v.s. existing methods that do not enforce monotonic alignment. Some studies [1] have shown that enforcing monotonic attention patterns can lead to much lower WER and even better naturalness (it is probably also the reason that it is proposed). This makes me question if the lower WER and higher MOS come primarily from the use of monotonic alignment, which is not the main novelty of the paper, rather than from the use of GMM-VAE and GMM-LM. Furthermore, while the authors do provide a comparison of alignment methods in Appendix A.1, the one without monotonic alignment (Cross Att.) does result in higher WER than all the baselines. This further substantiates that the performance increase may not come from GMM-VAE and GMM-LM. I would suggest the author do an ablation study on the monotonic alignment and add it to Table 2 to interleave the contributions of the two components.

[1] L. Chen, A. Rudnicky, S. Watanabe, ”A Vector Quantized Approach for Text to Speech Synthesis on Real-World Spontaneous Speech”, in Proceedings of AAAI 2023, 2023

**Questions:**

- Section 3.3 is a little bit unclear. For instance, how was this energy function $e_{ij}$ calculated? Does it have something to do with the cross-attention weights acquired by the transformer?
- What are the specs for the baseline methods: StyleTTS-2 and HierSpeech++? Are they of similar parameter size?
- I am wondering the validity of modeling the autoregressive distribution as a GMM. In your case, even if the KL regularization loss of GMM-VAE makes $q(h|x)$ a mixture of Gaussians, does it in any sense implies that the autoregressive conditional distribution $p(h_t|h_{t-1}, \cdots)$ is also close to a Gaussian mixture?

The paper is well-written and interesting, but I think the experiment issue mentioned in Weaknesses should be addressed before the paper is ready to be published.

---

> ### Author Response · Authors · 2024-11-20
>
> We sincerely thank the reviewer for taking the time to read our paper and provide thoughtful feedback. We have addressed the reviewer’s comments and concerns as outlined below.
>
> ### 1. Concerns regarding the separation of contributions between the GMM approach and the proposed monotonic alignment.
> We understand the reviewer’s concerns and explain below why the proposed monotonic alignment was not included in VALL-E and other models, along with our efforts to separate contributions in the revised manuscript.
>
> **Why the proposed monotonic alignment was not included in VALL-E and other models**: Our monotonic alignment method refines cross-attentions between the encoder and decoder to enforce monotonic alignment. However, it is unclear how to apply this method to a decoder-only model like VALL-E, which relies solely on self-attention. This is why we did not include the proposed monotonic alignment for VALL-E.
> As for StyleTTS2 and HierSpeech++, both already incorporate monotonic alignment mechanisms. StyleTTS2 uses a duration prediction model to enforce strict monotonic alignments, while HierSpeech++ employs monotonic alignment search (MAS), as introduced in the VITS paper.
>
>  **Is monotonic alignment the main or the only reason for WER improvement?**
> We agree with the reviewer that it is important to separate the contribution of continuous tokenization in our GMM approach from the proposed monotonic alignment method. To address this, we conducted a comparison using discrete autoregressive (AR) models with the proposed monotonic alignment method, implemented using the same architecture as the GMM-LM large version (315m), except for the discrete codec embedding layer and softmax layers.
>
> Specifically, we implemented two discrete AR encoder-decoder models:
>
> 1. The first model used discrete codes extracted from a VQ-VAE codec model, implemented with a single codebook containing 8192 entries and using the same architecture as the DAC model [1]. This model was trained to predict the next tokens using standard cross-entropy with a single softmax layer.
>
> 2. The second model used discrete codes extracted from the DAC model with 8 codebooks, each containing 1024 entries. The model adopted delayed codebook prediction with multiple softmax layers, as proposed in MusicGen [2] and [3].
>
> The WER (%) of these models are presented in Table 10 of the revised manuscript and summarized below.
> | **Model**                     | **Output**               | **Codec Model** | **Cross Att.** | **Mono. Align.** |
> |-------------------------------|--------------------------|-----------------|----------------|------------------|
> | Discrete AR                   | Logits Group            | VQ-VAE          | 8.02           | 5.35             |
> | Discrete AR with Delay Pred.  | Multiple Logits Groups  | DAC             | 7.83           | 5.87             |
> | GMM-LM                        | GMM Parameters          | GMM-VAE         | 6.60           | 2.72             |
>
> The results clearly show that monotonic alignment contributes to WER  improvement. However, even with monotonic alignment, the discrete models do not achieve the same performance as the proposed GMM-based version. We believe this performance gap can be attributed to two factors:
>
> 1. **Quantization and discrete representations introduce limitations:**
>    Discrete representations can result in mispronunciations and artifacts. This is why some researchers use embeddings before the AR model's softmax layer as input to waveform decoders as a workaround [4,5].
>
> 2. **Challenges of applying monotonic alignment to RVQ-based models:**
>    RVQ models increase the complexity of TTS systems, as they require multiple softmax heads to predict codes from different codebooks. While it is possible to predict all codebooks in parallel at each timestep, this approach ignores dependencies between codebooks and yields suboptimal results. A better approach, as used in VALL-E and the delayed prediction model in Table 10, is to predict "coarse" codes first, followed by "fine" codes. However, this approach complicates monotonic alignment because alignment must be performed using only the coarse codes at each timestep. This limitation likely contributes to the higher WER observed with RVQ-based models.
>
> [1] Kumar, Rithesh, et al. "High-fidelity audio compression with improved rvqgan." *Advances in Neural Information Processing Systems* 36 (2024).
>
> [2] Copet, Jade, et al. "Simple and controllable music generation." *Advances in Neural Information Processing Systems* 36 (2024).
>
> [3] Lyth, Dan, and Simon King. "Natural language guidance of high-fidelity text-to-speech with synthetic annotations." *arXiv preprint arXiv:2402.01912* (2024).
>
> [4] Casanova, Edresson, et al. "XTTS: a Massively Multilingual Zero-Shot Text-to-Speech Model." *arXiv preprint arXiv:2406.04904* (2024).
>
> [5] Betker, James. "Better speech synthesis through scaling." *arXiv preprint arXiv:2305.07243* (2023).

---

> ### Author Response · Authors · 2024-11-20
>
> ### 2. Section 3.3 is a little bit unclear. For instance, how was this energy function $e_{ij}$ calculated? Does it have something to do with the cross-attention weights acquired by the transformer?
>
> The energy is computed using cross-attention between encoder and decoder features, and monotonic alignment is enforced based on the energy terms. We have added Algorithm 1 in the appendix, which provides a detailed explanation of the energy terms, monotonic alignment process, and decoder functionality.
>
> ### 3. What are the specs for the baseline methods: StyleTTS-2 and HierSpeech++? Are they of similar parameter size?
>
> We used the standard versions of the models  in their  GitHub repositories. The StyleTTS-2 models have a total of 142M parameters, and HierSpeech++ has 97M parameters. Both models have more parameters than our GMM-LM mini model, which outperforms them in both MOS and WER. We have included the model size information in the revised manuscript.

---

> ### Author Response · Authors · 2024-11-20
>
> ### 4. I am wondering the validity of modeling the autoregressive distribution as a GMM. In your case, even if the KL regularization loss of GMM-VAE makes $q(h|x)$ a mixture of Gaussians, does it in any sense implies that the autoregressive conditional distribution $p(h_t|h_{t-1}.\ldots)$  is also close to a Gaussian mixture?
>
> The validity of interpreting the AR distribution as a GMM can be justified by the fact that GMMs are universal approximators of any smooth distribution [1,2]. Thus, the question is not whether a given distribution is inherently a GMM but rather whether it can be efficiently approximated by a GMM. Our positive results confirm that this is indeed achievable.
>
> Intuitively, the latent space distribution $ p(h) $ can be interpreted as a marginalization over all possible phonemes, while the conditional distribution $p(h_t | \text{phoneme}) $ represents the distribution conditioned on a particular phoneme. For a specific phoneme, $p(h | \text{phoneme})$ is likely a subset of the GMM. Since this exact subset is unknown, we train a model to predict it.
>
> [1] Carreira-Perpinan, Miguel A. "Mode-finding for mixtures of Gaussian distributions." *IEEE Transactions on Pattern Analysis and Machine Intelligence* 22.11 (2000): 1318-1323.
>
> [2] Goodfellow, I., Bengio, Y., & Courville, A. (2016). *Deep Learning*. MIT Press.

---

> > ### Author Response · Authors · 2024-11-20
> >
> > ### 5. Some studies [1] have shown that enforcing monotonic attention patterns can lead to much lower WER and even better naturalness (it is probably also the reason that it is proposed).
> >
> > Thank you for pointing this out. We have added the suggested reference in the revised manuscript to highlight that monotonic attention can improve model WER and naturalness.

---

> ### Comment · Reviewer_UrKc · 2024-11-20
>
> I thank the authors for the detailed experiments. This additional experiments of comparing with the Discrete Methods w. monotonic alignment from the authors resolves my main concern about the paper. I am willing to raise the score.
>
> Additionally, I want to clarify on the fifth point of the author's response, I provided that reference to provide grounding for why the the ablation of monotonic alignment is needed, not to suggest inadequate reference of the original submission.

---

> > ### Author Response · Authors · 2024-11-21
> >
> > We appreciate the reviewer’s kind words about our efforts and look forward to continuing the discussion if any additional questions arise.

---

### Official Review · Reviewer_dDJ8 · 2024-10-28

**Soundness:** 3
**Presentation:** 3
**Contribution:** 3
**Rating:** 8
**Confidence:** 5

**Summary:**

Authors propose a novel means of auto-regressive TTS modeling that eschews quantization units in favor of Gaussian mixtures. Model performance consistently outperforms other standard TTS models, demonstrating high quality TTS is able without traditional VQ-VAE setup.

**Strengths:**

Authors present provide a thorough discussion of related work and their motivation for their approach. Description of architecture is clear and easy to follow, along with pointers for reproducibility. High performance of model is significant enough for comparison with other approaches.

**Weaknesses:**

There is a minor question of motivation in the author's approach: they take the stance that the community views vector quantization approaches as a necessity, but there are a fair amount of approaches in the speech modeling community that have used straight reconstruction approaches. While their gaussian mixture approach is still suitably novel, this position seems to ignore other considerations that go into VQ approach. Notably that the use of discrete tokens is relatively easy to implement in parallel with text encoding, all while minimizing storage an I/O limitations from audio/image processing.

**Questions:**

Given the reliance of the model architecture on monte-carlo estimation, how sensitive are results to random seeding during expeirmentation?

What is the performance on more noisy datasets than LibriSpeech? Is the Gaussian approach suitably robust across evaluation sets?

---

> ### Author Response · Authors · 2024-11-20
>
> We sincerely thank the reviewer for taking the time to read our paper and provide thoughtful feedback. We have addressed the reviewer’s comments and concerns as outlined below.
>
> ### 1. There is a minor question of motivation in the author's approach: they take the stance that the community views vector quantization approaches as a necessity, but there are a fair amount of approaches in the speech modeling community that have used straight reconstruction approaches.
>
> Thanks for point this out. We have added several literatures using reconstruction and multi-task approaches for learning speech features as the reviewer suggested in literature review section (line 88-91).
>
> ### 2. While their gaussian mixture approach is still suitably novel, this position seems to ignore other considerations that go into VQ approach. Notably that the use of discrete tokens is relatively easy to implement in parallel with text encoding, all while minimizing storage an I/O limitations from audio/image processing.
>
> We have emphasized in the revised manuscript that the VQ approach is more efficient in terms of storage and I/O and enables the direct use of language models (line 100-103).
>
> ### 3. Given the reliance of the model architecture on monte-carlo estimation, how sensitive are results to random seeding during experimentations?
>
> We found that the model is not sensitive to random seeds in our experiments. With a batch size of 680 for GMM-VAE training, we observed no noticeable variations between runs. We have clarified this for readers in the revised manuscript.

---

> > ### Author Response · Authors · 2024-11-20
> >
> > ### 4. What is the performance on more noisy datasets than LibriSpeech? Is the Gaussian approach suitably robust across evaluation sets?
> >
> > We have added an experiment in Section A.4 of the appendix to discuss how the noise level of the prompt affects zero-shot TTS performance. The reviewer may refer to Table 8 in the revised manuscript or the table below for the results. In summary, the prompt noise level impacts the speaker similarity of the synthesized speech more than WER, and the GMM-LM demonstrates better preservation of speaker similarity across different noise levels.
> > | **Model**        | **WER (%) -20 dB** | **WER (%) -15 dB** | **WER (%) -10 dB** | **SIM -20 dB** | **SIM -15 dB** | **SIM -10 dB** |
> > |-------------------|--------------------|--------------------|--------------------|----------------|----------------|----------------|
> > | StyleTTS-2       | 3.45               | 3.49               | 3.54               | 0.72           | 0.71           | 0.68           |
> > | HierSpeech+++    | 3.61               | 3.72               | 3.91               | 0.69           | 0.64           | 0.58           |
> > | VALL-E           | 9.84               | 10.86              | 12.61              | 0.61           | 0.54           | 0.51           |
> > | Ours-Mini        | 2.85               | 3.01               | 3.04               | 0.77           | 0.74           | 0.74           |
> > | Ours-Large       | **2.77**           | **2.82**           | **2.97**           | **0.88**       | **0.87**       | **0.85**       |

---

### Official Review · Reviewer_8JGM · 2024-11-09

**Soundness:** 3
**Presentation:** 3
**Contribution:** 3
**Rating:** 6
**Confidence:** 4

**Summary:**

This paper presents a novel approach to autoregressive speech modeling using continuous speech features, contrast to recent trends that rely on discrete units. The method consists of two key components: (1) A feature extraction model based on VAE, which replaces quantized codebooks (as in RVQ) with a learned mixture of Gaussian priors (GMM-VAE), and (2) A text-to-speech model that employs a Gaussian Mixture Model Language Model (GMM-LM) to model these continuous features in autoregressive manner, which also incorporating a new monotonic alignment constraint. Experimental results demonstrate that this continuous speech modeling consistently outperforms previous methods using discrete codec representations like Residual Vector Quantization (RVQ) in TTS tasks.

**Strengths:**

- Clear comparison to previous approaches and introduces novel continuous variants for both VAE training and TTS stages
- The introduction of GMM-LM is novel, and the formulation is clear and simple. It also enables probabilistic sampling which is a plus for TTS applications
- Nice results with much less model parameters

**Weaknesses:**

- Limited discussion of prior Gaussian mixture VAE work, e.g., "Deep Unsupervised Clustering with Gaussian Mixture Variational Autoencoders". (Minor: The notation, either GMM-VAE or VAE-GMM, should be consistent.)
- The counterintuitive result where increasing Gaussian mixtures in GMM-VAE leads to worse reconstruction, where a 6-mixture GMM should subsume the modeling capacity of a 3-mixture GMM
- Some modeling details are missing, e.g., GMM-VAE frame rate, which is crucial as it could affect the type of information captured

**Questions:**

- Does the frame rate of GMM-VAE features align with the mel spectrogram hop length (240ms) described in 5.2?
- Why does the model require relatively few Gaussian mixtures compared to VQ codes in RVQ, and any insights on what the mixture components capture?
- Could you clarify how the monotonic alignment mechanism in Figure 2 (right) works? It seems to align the encoded text and speech prompts prior to decoding. Additionally, a more comprehensive description of the GMM-LM would be nice, including how speech and text features are fed into each decoder step, and the formulation of $e_{i,j}$​.

---

> ### Author Response · Authors · 2024-11-20
>
> We sincerely thank the reviewer for taking the time to read our paper and provide thoughtful feedback. We have addressed the reviewer’s comments and concerns as outlined below.
>
>
> ### 1.  Limited discussion of prior Gaussian mixture VAE work, e.g., "Deep Unsupervised Clustering with Gaussian Mixture Variational Autoencoders"(Minor: The notation, either GMM-VAE or VAE-GMM, should be consistent.).
>
> Thanks for pointing out the relevant prior work on Gaussian mixture VAEs. We have incorporated the suggested reference, along with several other works on VAEs with learned priors, into the literature review section under the subsection "VAEs with Learned Priors" (lines 146–155).  We have made the notation consistent as “GMM-VAE” throughout the manuscript, replacing the two instances of “VAE-GMM.
>
> ### 2. The counterintuitive result where increasing Gaussian mixtures in GMM-VAE leads to worse reconstruction, where a 6-mixture GMM should subsume the modeling capacity of a 3-mixture GMM
> The reconstruction loss reported in Table 4 is based on the test set (LibriSpeech test clean). The performance gap in terms of VAE reconstruction is relatively small (0.71 vs. 0.73 when λ=0.1, and for λ=10, the 6-mixture model slightly outperforms the 3-mixture model with 0.98 vs. 0.95). To investigate further, we sampled a subset of the training data (200 randomly selected speakers with 5 utterances each) and computed the reconstruction loss. In this case, the 6-mixture model consistently outperformed the 3-mixture model, suggesting that the 6-mixture model may slightly overfit the data. We have included these additional experiments (Table 9) and discussions in Section A.3 of the revised manuscript.
> | **No. Gaussian** | **Mode**         | **0.1** | **1** | **10** | **50** | **100** |
> |-------------------|------------------|---------|-------|--------|--------|---------|
> | 1                 | Training Set     | 0.72    | 0.78  | 0.98   | 1.54   | 1.85    |
> | 1                 | Evaluation Set   | 0.76    | 0.89  | 1.14   | 1.67   | 2.06    |
> | 3                 | Training Set     | 0.68    | 0.70  | 0.93   | 1.11   | 1.54    |
> | 3                 | Evaluation Set   | 0.71    | 0.77  | 0.98   | 1.13   | 1.77    |
> | 6                 | Training Set     | 0.67    | 0.73  | 0.86   | 1.02   | 1.32    |
> | 6                 | Evaluation Set   | 0.73    | 0.86  | 0.95   | 1.37   | 1.83    |
>
> Additionally, we believe that when the GMMs are parameterized using neural networks, the 6-mixture GMM may not fully subsume the modeling capacity of a 3-mixture GMM. In a traditional GMM, a 6-mixture model typically has twice as many parameters as a 3-mixture model, allowing it to subsume the latter’s modeling capacity. However, in our approach, which utilizes a neural network for parameterization, both the 3- and 6-mixture models have approximately the same total number of parameters. Consequently, the 6-mixture model has fewer parameters per mixture, which may limit its ability to completely subsume the modeling capacity of a 3-mixture GMM.
>
> ### 3. Some modeling details are missing, e.g., GMM-VAE frame rate, which is crucial as it could affect the type of information captured
>
> For GMM-VAE, we used the same model architecture as described in the DAC paper [1] with modifications that removed all quantization-related layers. The  encoder rate are decoder rate are [8, 8, 4, 2], for 16khz speech it is  320ms frame rate. We have included the frame rate and the more details of GMM-VAE in section A.5  of the revised paper.
>
> [1] Kumar, Rithesh, et al. "High-fidelity audio compression with improved rvqgan." *Advances in Neural Information Processing Systems* 36 (2024).
>
> ### 4. Does the frame rate of GMM-VAE features align with the mel spectrogram hop length (240ms) described in 5.2?
>
> They are not the same. The frame rate of GMM-VAE is 320. The mel spectrogram hop length is 240ms. We have included this detail in the revised paper.

---

> ### Author Response · Authors · 2024-11-20
>
> ### 5. Why does the model require relatively few Gaussian mixtures compared to VQ codes in RVQ.
>
> To explain this phenomenon, we consider the distinctions between Vector Quantization (VQ) codes and Gaussian Mixture Models (GMMs) in representing distributions. VQ codes do not directly correspond to Gaussian mixtures; instead, they represent a distribution similarly to fitting a histogram, where probability mass is assigned only to specific embedding points (the codebook vectors). This approach allows for the flexible placement of probability mass exactly where it is needed, unconstrained by the form of continuous distributions like GMMs, which always assign some probability mass around the modes and in the surrounding regions.
>
> VQ works particularly well for distributions with many modes and minimal probability mass between them. It efficiently captures each distinct mode without concern for intermediate regions. However, it can become inefficient if there is significant probability mass between the modes, as it would require a large number of codebook vectors to accurately represent the distribution.
>
> On the other hand, GMMs can model multi-modal distributions effectively by utilizing continuous distributions. They naturally allocate probability mass not only at the modes but also between them, due to the inherent properties of Gaussian functions. If the modes are far apart, a GMM may need as many components as there are modes, potentially making it less efficient than the VQ approach in such scenarios. However, for distributions where the modes are close and there is a smooth transition between them, GMMs are better suited because they can capture the gradual changes in probability density effectively.
>
> Therefore, the choice between VQ and GMM depends on the nature of the latent space distribution. For distributions with distant modes and minimal probability mass between them, VQ or other discrete distributions are more effective. Conversely, for distributions where the modes are close and there is significant probability mass in between, continuous distributions like GMMs are preferable. Based on our experiments, it appears that the latter scenario applies in this case, indicating that GMMs are the more appropriate choice.

---

> ### Author Response · Authors · 2024-11-20
>
> ### 6. Could you clarify how the monotonic alignment mechanism in Figure 2 (right) works? It seems to align the encoded text and speech prompts prior to decoding. Additionally, a more comprehensive description of the GMM-LM would be nice, including how speech and text features are fed into each decoder step, and the formulation of $e_{i,j}$.
>
> In the revised manuscript, we have provided a step-by-step explanation in Algorithm 1 detailing how text and speech features are aligned monotonically, integrated into the decoder, and used to compute the negative log-likelihood.

---

### Author Response · Authors · 2024-11-20

We sincerely thank the reviewer for taking the time to read our paper and provide thoughtful feedback, which has significantly improved the manuscript. Based on the feedback, we have made the following revisions:

1. **Detailed Description of the Proposed Monotonic Alignment and the Decoder:**  We included Algorithm 1, providing a step-by-step explanation of the proposed monotonic alignment process and the operation of the GMM-LM decoder.

2. **Experiments with Proposed Monotonic Alignment in Discrete Models:**  We conducted additional experiments to evaluate the performance of discrete AR encoder-decoder models with the proposed monotonic alignment.

3. **Robustness to Prompt Noise Experiments:**  We investigated the robustness of the proposed GMM-LM against noise in prompts and included the findings in the revised manuscript.

4. **GMM-VAE Reconstruction Analysis:**  We analyzed why the GMM-VAE with 6 mixtures underperforms compared to 3 mixtures in terms of reconstruction quality and provided an explanation in the revised manuscript.

5. **References:**  We added more references, highlighting the benefits of monotonic alignments, VAEs with learned priors, and advancements in speech feature learning.

6. **Model Details:**  As suggested, we included additional details about both the GMM-VAE and baseline models to enhance clarity and completeness.

---

### Meta-Review · Area_Chair_dt3G · 2024-12-21

**Metareview:**

The paper proposes probabilistic improvements to sequence-to-sequence TTS systems. Residual quantization is replaced by GMM-VAE, and next-token prediction is also modeled by a GMM. The experiments focus on zero-shot TTS, where the proposed approach is compared to StyleTTS-2, VALL-E, and HierSpeech++. Results show that the proposed approach perform well on both objective and subjective evaluations.

All reviewers found the approach refreshing and were satisfied with the empirical justification. There were a few minor questions. The discussion was healthy, and the reviewers were happy with the revision.

**Additional Comments On Reviewer Discussion:**

There were a few technical questions, such as whether the performance corresponds well with the number of mixture components, how the noise affects the synthesis, and the decoupling of monotonic alignments and GMM. The authors responded well and were able to resolve issues by providing further evidence.

---

### Decision · Program_Chairs · 2025-01-22

Accept (Poster)